# Active Pulmonary Tuberculosis Triggered by Interferon Beta-1b Therapy of Multiple Sclerosis: Four Case Reports and a Literature Review

**DOI:** 10.3390/medicina56040202

**Published:** 2020-04-24

**Authors:** Carmen Adella Sirbu, Elena Dantes, Cristina Florentina Plesa, Any Docu Axelerad, Minerva Claudia Ghinescu

**Affiliations:** 1Neurology Department, Titu Maiorescu University, 040441 Bucharest, Romania; sircar13@yahoo.com (C.A.S.); plesacristina@yahoo.com (C.F.P.); ghinescu_minerva@yahoo.com (M.C.G.); 2Neurology, Carol Davila Central Military Emergency University Hospital, 010242 Bucharest, Romania; 3Department of Pneumology, Faculty of Medicine, Clinical Hospital of Pneumophtisiology, ‘Ovidius’ University of Constanta, 900527 Constanta, Romania; 4Department of Neurology, Faculty of Medicine, Clinical Emergency Hospital ‘St. Apostol Andrei’, ‘Ovidius’ University of Constanta, 900527 Constanta, Romania; docuaxi@yahoo.com

**Keywords:** IFNβ-1b, multiple sclerosis, latent infection, *Mycobacterium tuberculosis*, active tuberculosis, comorbidities

## Abstract

In this paper, we reported on four cases of severe pulmonary active tuberculosis in patients with multiple sclerosis (MS) undergoing interferon beta-1b (IFNβ-1b) therapy. Disease-modifying therapies (DMTs) in MS may increase the risk of developing active tuberculosis (TB) due to their impact on cellular immunity. Screening for latent infection with *Mycobacterium tuberculosis* (LTBI) should be performed, not only for the newer DMTs (alemtuzumab, ocrelizumab) but also for IFNβ-1b, alongside better supervision of these patients.

## 1. Introduction

Multiple sclerosis (MS) is a chronic, neurodegenerative, demyelinating and inflammatory disease which may be accompanied by other autoimmune or infectious diseases due to its immunological peculiarities and the immunomodulatory nature of the treatment applied to patients [1]. Given their impact on cellular immunity, disease-modifying therapies (DMTs) used to treat MS could increase the risk of infections. Global surveillance, albeit with a limited follow-up period, has previously found that highly effective DMTs (fingolimod, natalizumab) are associated with an increased risk of infections compared to placebo or first line therapies (interferon beta and glatiramer acetate) [2]. A recent Swedish study found an increased risk of infections of approximately 50% among people with MS who were treated with first line DMTs compared to the general population [3]. Another randomized clinical trial of the recently approved anti-CD20 therapy for MS, ocrelizumab, reported an increased risk of respiratory tract infections compared with the interferon beta treatment and placebo [4,5]. The risk of developing tuberculosis (TB) due to MS and DMTs is unknown. Moreover, we did not find any reported cases of active TB disease in patients with multiple sclerosis undergoing interferon beta-1b (IFNβ-1b) treatment. We found a single case of an MS patient treated with glatiramer acetate who had contact with infected persons, and who underwent specific treatment for nine months, for whom TB was reactivated after 16 years [6]. The risk of TB has been reported only in patients treated with newer DMT drugs, such as alemtuzumab, natalizumab, fingolimod, mitoxantrone and dimethyl fumarate [7]. In clinical trials, very few TB cases have been reported during treatment with cladribine and teriflunomide [8]. There is over 25 years of experience regarding the benefits, safety and tolerability of IFNβ-1b treatment, as this was the first DMT used for the treatment of MS, beginning in 1993. IFNβ-1b binds to type I interferon receptors. Approximately 100 immunomodulatory and antiviral proteins are activated by their phosphorylation and dimerization. Chronic subcutaneous administration every other day, at the recommended dose of 0.25 mg (1 mL), has been shown to induce long-term effects in the expression of neuroprotective genes, brain repair and clinical efficacy [9]. However, the exact mechanism of action is not known. Tuberculosis screening is recommended for patients with multiple sclerosis before starting with certain therapies that modify the disease, which does not include IFNβ-1b. DMTs may affect interferon-gamma release test results. The QuantiFERON-TB Gold test measures the response of T cells stimulated by *Mycobacterium tuberculosis* (Mtb) antigens [10].

## 2. Case Reports

We presented four patients diagnosed with relapsing-remitting MS (RRMS) and secondary progressive MS (SPMS) who developed pulmonary TB while on chronic IFNβ-1b treatment. They were selected from the patients diagnosed and treated at our center, which has 20 years of experience based on the care of over 400 MS patients. The demographic, diagnostic and treatment data, as well as the particularities of each case, are listed in Table 1.

None of these cases had any known contact with active TB patients, other immunodepressive states, or risk factors, except for a history of pulse corticosteroid therapy with methylprednisolone. All were HIV negative. Other causes of pulmonary involvement were excluded. When active TB was confirmed, IFNβ-1b treatment was discontinued and resumed immediately after the cessation of TB treatment. In all cases, after being clinically and visually detected, TB was bacteriologically confirmed by a microscopic examination of Ziehl–Neelsen-stained smears and culture in a solid Löwenstein–Jensen medium. TB treatment was administered according to the National Program for Tuberculosis Prevention, Monitoring, and Control. All patients had advanced forms of drug-susceptible secondary (post-primary) tuberculosis, which occurred after at least one year of IFNβ-1b treatment. All cases presented ulcerated caseous lesions that spread to at least one pulmonary lobe and two cases were highly contagious forms, with the sputum smear microscopy results being positive. In all cases, the initial indication was the standard six-month regimen consisting of isoniazid (H), rifampicin (R), pyrazinamide (Z), and streptomycin (S) or ethambutol (E) given daily (7/7) for the first two months, followed by four months of isoniazid and rifampicin given three times a week (2HRZS (E) 7/7/4HT3/7). The treatment was prolonged to eight months due to lesion extension (cavitary lesions) and 12 months due to an individualized regimen (without H and E). TB relapse occurred in only one case after 14 years. All the cases were bacteriologically monitored and were considered cured. One of the patients developed a bladder tumor 17 years after the IFNβ-1b treatment. All patients developed different comorbidities over the course of their MS.

## 3. Discussion

Active tuberculosis has not yet been reported as an infectious complication of IFNβ-1b therapy with MS patients [7]. IFNβ-1b decreases the expression of Very Late Antigen-4 (VLA-4) adhesion molecules and the penetration of activated lymphocytes through the blood–brain barrier, with a role in gene regulation [11]. Recent studies on MS pathophysiology have shown an imbalance of adaptive immunity (T regulator, B cell, cytokine, monocytes, Th1, Th17, Th2 and proinflammatory products) due to abnormalities in over 8000 expressed genes that control immune regulation, including interferon (IFN) signaling. Although the mechanisms are diverse and very complex, it seems that dysregulation of the IFN pathway is associated with active forms of MS and poor prognosis [9]. Pulmonary involvement in MS may occur as a result of disease progression to respiratory failure, secondary to respiratory musculature weakness and abnormalities in the neural control of respiration, but also as a result of the complications secondary to the treatment of the underlying disease [12]. However, defense against TB requires a complex range of innate (early response) and adaptive immune mechanisms involving a variety of immune cells and cytokines. In active TB, severity correlates with the circulating IFN-gamma levels, which play a special role in the activation of myeloid cells and in the inhibition of bacterial replication Tzelepis [13]. There are no data regarding a possible link between IFNβ-1b treatment in patients with multiple sclerosis and active tuberculosis. This is why we considered it important to present these cases. They may encourage more extensive research that will lead to new data in this field. Despite the different results of type I IFN response to infection, it is well documented that many intracellular, non-viral pathogens elicit a host response that results in an increased IFN-beta production [14]. There are different signaling pathways defining different gene expressions during active tuberculosis infection. However, the mechanism by which *Mycobacterium tuberculosis* infection regulates interferon-stimulated genes in human macrophages remains unknown [15]. Researchers at the Max Planck Institute (MPI) have patented a host cell model that functionally reproduces pulmonary alveolar macrophages (AM). Thus, the host interactions under infection with Mtb can be studied in vitro. The innate primary immune response of MPI cells in the presence of Mtb showed a large and early induction of the pro-inflammatory cytokines Tumor Necrosis Factor Alpha (TNFα), interleukin 6 (IL-6), IL-1α, and IL-1β, and elimination of the bacterium by phagolysosomes [16]. Studies have found that vitamin B5 can stimulate epithelial cells to express proinflammatory and antibacterial cytokines in macrophages infected with *Mycobacterium tuberculosis* [17].

Some Toll-like receptors on the surfaces of immune cells can identify bacteria, playing critical roles in tuberculosis infection. The receptors 2, 4, and 9 also have a fundamental role in pathology, with their expression levels being increased in MS. Receptors such as peptidoglycan, a major component of mycobacterial cell walls, have been identified in the CNS endothelial cells, cerebrospinal fluid (CSF) and glial cells of MS patients. These receptors are crucial in the primary identification of Mtb and the proper development of immune responses to overcome the infection [18]. The microbial agent activates the recognition receptors by initiating the innate immune response. *Mycobacterium tuberculosis* RNA determines, through the SecA2 and ESX-1 secretion system, the production of interferon-beta. Until recently, these mechanisms were only known to occur in infections with viruses [19].

In childhood, our studied patients received the Bacillus Calmette–Guérin (BCG) vaccine, which is part of the national mandatory vaccination program. BCG influences the transition from oxidative phosphorylation at aerobic glycolysis, thus ensuring the stimulation of the immunomodulation immune response and attenuation of mycobacterial infection. It has been shown that after BCG infection, IFN-β can enhance antigen-presenting cell (APC) activity, and the link between the innate and the adaptive immune systems. This could be an opportunity to find more effective vaccines in the fight against tuberculosis [20]. However, the protection lasts for only five to ten years after vaccination and therefore, scientists are looking for ways to improve and increase the vaccine’s efficiency [21,22]. We can speculate that IFNβ-1b therapy may impair protective immunity to *Mycobacterium tuberculosis*, given the complex immune mechanisms and genetic determinants of the two conditions. Neither the effect of interferons on humoral or cellular immunity, as is the case with other DMTs, nor the risk of opportunistic infections such as TB, are well known. The risk of TB depends on host susceptibility during exposure, and on their belonging to risk groups [23]. The risk of TB has not been described in MS patients on IFNβ-1b treatment. From recent data, the risks of tuberculosis reactivation in patients treated with alemtuzumab or teriflunomide is the highest and the recommendation is to screen for latent infection before starting therapy. For natalizumab, fingolimod, dimethyl fumarate and mitoxantrone, screening is optional. For monoclonal antibodies targeting CD20, the risk of reactivation is the lowest because B-cell depletion does not affect cell-mediated immunity [7]. A group of experts, neurologists, and pulmonologists have underlined that screening for latent tuberculosis infection (LTBI) is not required for MS patients treated with IFNβ-1b in low-prevalence countries (TB notification rate of <100 TB cases per million population per year) [24]. In our center, no patient was screened for LTBI before the initiation of the treatment for MS, although our country had a TB global incidence rate of 68 cases/100,000 persons per year during 2018. Screening for LTBI using interferon-gamma release assays (IGRAs) is considered necessary, even in low TB-endemic countries. The higher sensitivity and specificity of IGRAs have been reported, thus replacing the tuberculin skin test, especially in countries where the population is BCG-vaccinated [23]. However, certain treatments (fingolimod, dimethyl fumarate, methylprednisolone) may cause false-negative or indeterminate IGRA results. The effect of IFNβ-1b on IGRA results is not exactly known [25]. There is no test predictive of the progression from LTBI to active disease, which is the reason prevention plays an important role. Screening for LTBI is currently recommended before starting daclizumab and alemtuzumab therapy, but not before initiating CD20-acting DMTs (rituximab, ocrelizumab) [26,27]. Additionally, the screening of high-risk groups in the population and the use of infection control measures or chemoprophylaxis are indicated [23]. In the future, however, genetic tests could be used; to date, 16 genes with predictive values for susceptibility to developing TB have been described [28].

## 4. Conclusions

The presented cases demonstrated the occurrence of active tuberculosis after at least one year of IFNβ-1b treatment in young MS patients. It is the first report in the literature of this comorbidity involving IFNβ-1b treatment. The immunological changes in MS and TB appear to be extremely complex. The relationship between MS, IFNβ-1b and the drugs with the longest and most widespread use for these patients and active TB requires further clinical studies to quantify this risk. This is important, especially since the treatment of patients with MS is chronic, spanning numerous decades. Such research would help in developing guidelines around the risk of infections, including TB, in MS patients. The appearance of other comorbidities increases the burden of the patient’s suffering and implies the need for a better supervision of DMT-treated patients. Pre-treatment and annual screening for latent TB infection and the control of respiratory symptoms should be compulsory in any patient with MS when TB is endemic.

## Figures and Tables

**Figure 1 medicina-56-00202-f001:**
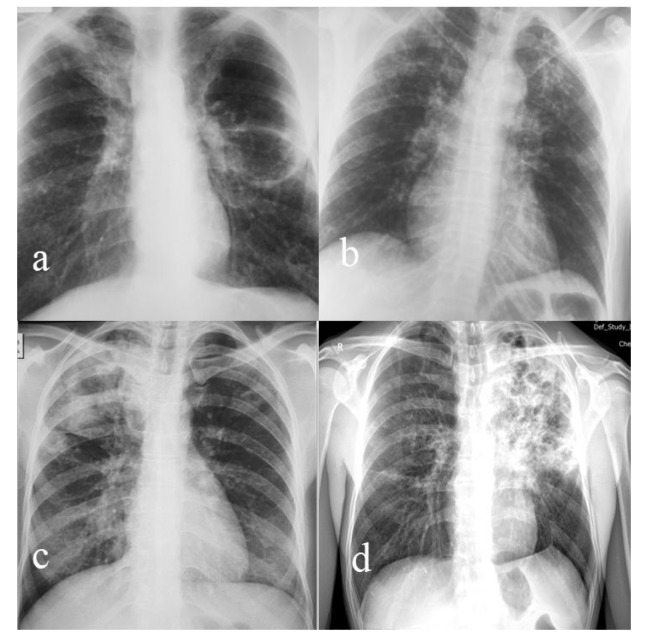
(**a**) Large fibrocavitary lesion of the Fowler segment with nodular bilateral bronchogenic disseminations on the lower lobes (case 1); (**b**) bilateral infiltrative nodular TB lesions upper lobes (case 2); (**c**) caseous ulcerated lesions, right upper lobe, with homo and contralateral nodular dissemination (case 3); (**d**) extensive caseous ulcerated TB left upper lobe with contralateral bronchogenic dissemination (case 4).

**Table 1 medicina-56-00202-t001:** Patient characteristics.

Characteristics	Case 1	Case 2	Case 3	Case 4
Gender	F	M	F	F
MS diagnosis form	26, RRMS	32, SPMS	30, RRMS	37, RRMS
Age of IFNβ-1b treatment initiation	27	45	32	39
MRI brain scan	2001, 2002, 2003, 2009, 2011, 2019	1998, 2000, 2007	2014, 2015, 2019	2006, 2008, 2015, 2018
EDSS progression	1 (2001); 5.5 (2019)	**3.5** (2000); **5.5** (2019);	**1** (2015); **4** (2019)	**1** (2007); **1.5** (2019)
Time between IFNβ-1b treatment initiation and TB	12 months	48 months	36 months	84 months
Treatments other than IFNβ-1b before TB	Methylprednisolone, antidepressants	Methylprednisolone, beta-blocker, antihypertensive, statins, antidepressants	Methylprednisolone	Methylprednisolone
Onset of active TB,	28 y	49 y	35 y	46 y
Onset of symptoms prior to TB	6 w	3 w	3 w	4 w
Respiratory symptoms	Hemoptysis, asthenia, weight and appetite loss, a sharp twinge in left side of chest	Asthenia, loss of appetite, productive cough, exertional dyspnea	A twinge in the left side of chest, weight loss	Asthenia, fever, persistent nonproductive cough
Radiological presentation (Figure 1)	Nodular and caseous cavitary lesions	Bilateral apical ulcerated fibrocaseous lesions with bronchogenic disseminations	Caseous-cavitary lesions	Infiltrative nodular and bronchial forms
TB localization	Apical segment (Fowler) left lower lobe	Bilateral upper lobes	Right upper lobe	Left upper lobe
Initial bacteriological examination of sputum	Negative AFB smear; positive for *M. tuberculosis* culture	Positive smear and culture	Positive smear and culture	Negative smear, positive culture
Category of the treatment regimen	2HRZS (5/7) +4HR (3/7)	2HRZE (7/7) +4HR (3/7)	2HRZE (7/7) + 6HR (3/7)	3HRZOfx (7/7) + 1HROfx (7/7) +8 OfxPr (3/7) (H-intolerance)
Treatment duration (months)	6	6	8	12
Bacteriological follow up after treatment initiation	Negative at 2, 4, 6 months	Negative at 2, 4, 6 months	Negative at 2, 4, 6, 8 months	Negative at 2, 4, 6, 8 months
Chest X-ray after DOT treatment	Left fibronodular sequelae	Right post-TB fibronodular sequelae	Right post-TB fibronodular sequelae	Several left subclavicular fibromicronodular lesions
Comorbidities	Anxiety–depressive disorder, osteopenia, vitamin D deficiency, *Escherichia Coli* urinary tract infection	Neurocognitive disorder, arterial hypertension, dyslipidemia, vitamin D deficiency	Depressive disorder, urinary incontinence	Depressive disorder, vitamin D deficiency
Special considerations and particularities of the case	Recurrence with extensive caseous–cavitary TB lesions left lung 14 years later. Positive sputum smear and culture. After 11 months of Category II treatment regimen, the bacteriological follow ups at 1, 3, 5, 8, and 11 months were negative (evaluated as cured).	Bladder tumor (invasive papillary urothelial carcinoma T2bN0M0 Grade III) Left ureterohydronephrosis	-	In 1996, recurrent transient vision loss, labeled retrobulbar optic neuritis, treated for 1 year with oral corticosteroids. Received an individualized treatment regimen due to isoniazid intolerance and exclusion of E due to its ocular side effects.

AFB- Acid-Fast Bacillus; F—female; M—male; IFNβ-1b—interferon beta 1 b; H―isoniazid 5mg/body weight; R―rifampin 10 mg/body weight; Z―pyrazinamide 30 mg/body weight; S―streptomycin 15 mg/body weight; E―ethambutol 25 mg/body weight; Ofx―ofloxacinum; Pr―protionamide 20 mg/body weight; 7/7—daily, 3/7—three times per week; 5/7- treatment administered five days per week; TB—tuberculosis; DOT- Directly Observed Therapy; RRMS—relapsing-remitting multiple sclerosis; SPMS—secondary progressive multiple sclerosis; EDSS—expanded disability status scale.

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
