# Peer review of "Active Pulmonary Tuberculosis Triggered by Interferon Beta-1b Therapy of Multiple Sclerosis: Four Case Reports and a Literature Review"

_medicina, 2020, doi:10.3390/medicina56040202_

Round 1
Reviewer 1 Report
The authors of the reviewed manuscript have reported the four cases of active tuberculosis (TB) in multiple sclerosis (MS) patients treated with interferon beta-1b (IFNβ-1b). In conditions that affect the immune system, such as MS, latent tuberculosis may thrive and reactivate during the use of immunomodulatory and immunosuppressive drugs. TB has been reported in patients treated with some drugs; among them alemtuzumab; thus, before the initiation of therapy, all patients should be evaluated for both, active and latent TB infection and treated according to local guidelines if required. However, there is not much information about possible correlation between IFNβ-1b treated patients with MS and active TB. That is why I found this “case report” interesting and important to evaluate more similar cases in future. I think that the presented subject is very important and falls into the profile of Medicina journal.
There are minor mistakes such as (but not limited to):
Abstract (line 18): please remove an extra dot;
Introduction (line 23): please add a full name and then abbreviation: disease-modifying therapies (DMTs);
Discussion (line 78): I do not know what you mean by “rate of 68% 000 population in 2018”, please correct this.
I suggest you re-write your introduction since there is insufficient information about what is already known about IFNβ-1b treated patients with MS and active TB. I think you should cite more papers, including a review published recently in Frontiers in Immunology by Rommer P. S. et al. (https://doi.org/10.3389/fimmu.2019.01564).
Also, please read carefully the whole manuscript and correct the other eventual errors.
Reviewer 2 Report
The authors provide a report of only four cases that they detected in patients undergoing therapy for MS. They do not follow up with all other patients which showed no signs of TB, checking these patients for Latent tuberculosis infection, immune status both pre and post treatment inititation etc could be a start.
The entire manuscript is nothing but a report that four such patients were found. I would like that the authors do a pubmed search and such papers are abundant in litterature.
In order for the manuscript to be worthy of publication I would like the authors to perform a more meaningful and worthy study looking into the affects of these drugs both pre and post the termination of MS treatments, possibly co-relating it with some more in-vito experiments.
Reviewer 3 Report
Here authors report four cases of severe pulmonary active tuberculosis in patients with multiple sclerosis (MS) and interferon 15 beta-1b (IFNβ-1b) therapy. They should provide details as suggested below. This could help in more direct understanding of cause and effect.
They should perform an in-vitro assay of macrophage infection with Mtb and treat with IFNb-1b and analyze for intracellular survival of Mtb.
They could also look into the induction of pro and anti-inflammatory cytokines in the IFNb-1b treated macrophage cells.
Authors should verify the line no 69 and 70 “in which IFN gamma plays a special role in the activation of myeloid cells in CD4 and CD8 and inhibition of bacterial replication” I did not understand what does this means.
Round 2
Reviewer 2 Report
The authors reply to the comments are satisfactory the manuscript may be published as a Case report.
The authors mention interesting studies and should consider making these a part of the manuscript
Author Response
Dear Reviewer,
Thank you very much for your efforts. Due to your comments, the quality of the article we hope has improved and now, meets your requirements! We modified the introduction, the discussions, and the conclusions.
Reviewer 3 Report
The article doesn't include any information other than as a reporting case, which has no scientific mechanism or detailed experiments. Authors commented on some of the answers about the mechanism but no experiments. Therefore, the article is not sufficient for the publication with current data.
Author Response
Response to Reviewer 3-round 2:
Dear Reviewer,
Thank you very much for your efforts. Due to your comments, the quality of the article we hope has improved and now, meets your requirements.
Experimental:
Researchers at the Max Planck Institute (MPI) have patented a host cell model that functionally reproduces pulmonary alveolar macrophages (AM). Thus, the host interactions - infection with Mycobacterium tuberculosis (Mtb) can be studied in vitro. The innate primary immune response of MPI cells in the presence of mycobacterium tuberculosis (Mtb) showed a large and early induction of pro-inflammatory cytokines TNFα, interleukin 6 (IL-6), IL-1α and IL-1β, and elimination of the bacterium by phagolysosomes. Woo M, Wood C, Kwon D, Park KH, Fejer G, and Delorme V (2018) Mycobacterium tuberculosis Infection and Innate Responses in a New Model of Lung Alveolar Macrophages. Front. Immunol. 9:438. doi: 10.3389/fimmu.2018.00438. Studies have found that vitamin B5 can stimulate epithelial cells to express proinflammatory and antibacterial cytokines in macrophages infected with Mycobacterium tuberculosis. He W, Hu S, Du X, Wen Q, Zhong X-P, Zhou X, Zhou C, Xiong W, Gao Y, Zhang S, Wang R, Yang J and Ma L (2018) Vitamin B5 Reduces Bacterial Growth via Regulating Innate Immunity and Adaptive Immunity in Mice Infected with Mycobacterium tuberculosis. Front. Immunol. 9:365. doi: 10.3389/fimmu.2018.00365.
Some Toll-like receptors on the surface of immune cells identifies bacteria, playing critical roles in tuberculosis. The receptors 2, 4 and 9, also play a fundamental role in pathology, their expression levels being increased in MS. Receptors such as peptidoglycan, a major component of mycobacterial cell walls, have been identified in CNS endothelial cells, cerebrospinal fluid (CSF), and glial cells of MS patients. These receptors are crucial in the primary identification of mycobacterium tuberculosis and proper development of immune responses to overcome infection. Faridgohar M, Nikoueinejad H. New findings of Toll-like receptors involved in Mycobacterium tuberculosis infection. Pathog Glob Health. 2017;111(5):256–264. doi:10.1080/20477724.2017.1351080.
The risk of a TB disease due to MS and DMTs is unknown. Moreover, we did not find any case of the TB active disease in patients with multiple sclerosis and IFNβ-1b treatment reported in the literature, yet (PubMed, Embase, EBSCO, MEDLINE, Web of Science Core Collection, and Google Scholar). We found a single case with MS on glatiramer acetate which had contact with infected persons, made specific treatment for 9 months and after 16 years TB was reactivated. Sanchez-Salcedo, P., & de-Torres, J. P. (2015). Immunomodulating Effects of Glatiramer Acetate and Its Potential Role in Pulmonary Tuberculosis Reactivation. Archivos de Bronconeumología (English Edition), 51(12), 656–657. doi:10.1016/j.arbr.2015.05.009.
The risk of tuberculosis (TB) has been reported only in patients treated with newer DMTs drugs, as alemtuzumab, natalizumab, fingolimod, mitoxantrone, and dimethyl fumarate. In clinical trials, very few TB cases were reported during treatments with cladribine and teriflunomide.
There are over 25 years of experience regarding the benefits, safety, and tolerability of interferon beta-1b (IFNβ-1b) treatment, being the first DMT used for the treatment of MS. IFNβ-1b decreases the expression of VLA -4 adhesion molecules, and the penetration of activated lymphocytes through the blood-brain barrier, with a role in gene regulation. Jakimovski D, Kolb C, Ramanathan M, Zivadinov R, Weinstock-Guttman B. Interferon β for Multiple Sclerosis. Cold Spring Harb Perspect Med. 2018;8(11):a032003. Published 2018 Nov 1. doi:10.1101/cshperspect.a032003.
IFNβ-1b binds to type I interferon receptors. Approximately 100 immunomodulatory and antiviral proteins are activated by their phosphorylation and dimerization. Chronic subcutaneous administration every other day of the recommended dose of 0.25 mg (1 mL), has been shown to induce long-term effects, correcting the dysregulated immune pathways with brain repair and good clinical response. But the exact mechanism of action is not known. Tuberculosis screening is recommended in patients with multiple sclerosis starting with certain therapies that modify the disease, but not for IFNβ-1b. DMTs may affect interferon-gamma release test results. The QuantiFERON-TB Gold test measures the response of T cells stimulated by Mycobacterium tuberculosis antigens. Pai M, Behr M. Latent Mycobacterium tuberculosis Infection and Interferon-Gamma Release Assays. Microbiol Spectr. 2016;4(5):10.1128/microbiolspec.TBTB2-0023-2016. doi:10.1128/microbiolspec.TBTB2-0023-2016
Global surveillance, but with limited follow-up period, found that highly effective DMT (fingolimod, natalizumab) were associated with an increased risk of infections compared to placebo or first-line therapy ( interferon beta and glatiramer acetate). Grebenciucova E, Pruitt A. Infections in patients receiving multiple sclerosis disease-modifying therapies. Curr Neurol Neurosci Rep. 2017;17(11):88. doi:10.1007/s11910-017-0800-8.
A recent Swedish study found an increase in infection risk of approximately 50% among people with MS who were treated with first-line DMTs therapies, compared to the general population. Luna, G., Alping, P., Burman, J., Fink, K., Fogdell-Hahn, A., Gunnarsson, M., … Frisell, T. (2019). Infection Risks Among Patients With Multiple Sclerosis Treated With Fingolimod, Natalizumab, Rituximab, and Injectable Therapies. JAMA Neurology. doi:10.1001/jamaneurol.2019.3365.
Another randomized clinical trial of recently approved anti-CD20 therapy for MS, ocrelizumab, reported an increased risk of respiratory tract infections compared with interferon beta treatment and placebo. Hauser SL, Bar-Or A, Comi G, et al; OPERA I and OPERA II Clinical Investigators. Ocrelizumab versus interferon beta-1a in relapsing multiple sclerosis. N Engl J Med. 2017;376(3):221-234. doi:10.1056/ NEJMoa1601277. Montalban X, Hauser SL, Kappos L, et al; ORATORIO Clinical Investigators. Ocrelizumab versus placebo in primary progressive multiple sclerosis. N Engl J Med. 2017;376(3):209-220. doi: 10.1056/NEJMoa1606468.
These presented cases demonstrate the occurrence of active tuberculosis after at least one year of IFNβ-1b treatment in young MS patients. It is the first report in the literature of this comorbidity, regarding IFNβ-1b treatment. Immunological changes in MS and TB appear to be extremely complex. The relationship between MS, IFNβ-1b, the drug with the longest and most widespread use for these patients and active TB, requires clinical studies to quantify this risk. This is important, especially since the treatment of patients with MS is a chronic one, over decades. These would help in developing guidelines for the risk of infections, including TB, in MS patients. The appearance of other comorbidities increases the burden of the patient's suffering and implies the need for better supervision of DMTs-treated patients. Pre-treatment and annual screening for latent TB infection and control of respiratory symptoms should be compulsory in any patient with MS when TB is endemic above the average.